# Antibiotic Prescribing Practices in Endodontic Infections: A Survey of Dentists in Serbia

**DOI:** 10.3390/antibiotics10010067

**Published:** 2021-01-12

**Authors:** Milan Drobac, Katarina Otasevic, Bojana Ramic, Milica Cvjeticanin, Igor Stojanac, Ljubomir Petrovic

**Affiliations:** Department of Dental Medicine, Faculty of Medicine, University of Novi Sad, Hajduk Veljkova 12, 21000 Novi Sad, Serbia; katarina.otasevic@mf.uns.ac.rs (K.O.); bojana.ramic@mf.uns.ac.rs (B.R.); milica.premovic@mf.uns.ac.rs (M.C.); igor.stojanac@mf.uns.ac.rs (I.S.); petns@uns.ac.rs (L.P.)

**Keywords:** antibiotic prescription, cross-sectional study, endodontic infections, Serbia

## Abstract

The study goal was to provide an overview of antibiotic prescribing practices of Serbian dentists when treating endodontic infections and to disseminate the current ESE (European Society of Endodontology) recommendations to the study participants. A link to an online questionnaire was sent to 628 Serbian dentists whose email addresses were publicly available on the Internet, 158 of whom responded to the survey, resulting in a 25.16% response rate. The significance of possible associations was assessed via the Chi-squared test and Cramer’s V measure of association, with *p* < 0.05 considered as statistically significant. According to the study findings, 55.7% of respondents prescribed a 5-day antibiotic course. Moreover, Amoxicillin 500 mg was the first-choice antibiotic for 55.1% of the respondents, followed by Clindamycin 600 mg (18.4%). For patients allergic to penicillin, 61.4% of respondents prescribed Clindamycin. Statistically significant differences emerged only in relation to acute apical abscess with systemic involvement, whereby dentists aged 46–55 were least likely to prescribe antibiotics in these clinical situations (*p* = 0.04). Analyses further revealed that recommendations for safe antibiotic prescribing practices were not always followed, as in certain cases, patients were given antibiotics even when this was not indicated. These findings highlight the need for additional education on responsible antibiotic use to prevent bacterial resistance.

## 1. Introduction

Excessive antibiotic use and consequent bacterial resistance are significant global problems [1]. Given that dentists prescribe approximately 10% of the antibiotics distributed in primary care, it is essential that they do so responsibly [2]. In order to prevent antibiotic overuse, in 2018, the European Society of Endodontology (ESE) issued the most recent recommendations for prescribing practices related to endodontic infections [3] and suggested that their members forward this information to dentists in their respective countries.

It is well known that antibiotics are ineffective in reducing pain or swelling of odontogenic origin in the absence of systemic signs of infection [2]. An ample body of empirical evidence indicates that antibiotics should only be prescribed to patients exhibiting systemic signs of infection or as a part of therapy offered to immunocompromised individuals [4,5,6]. In particular, they are not indicated for irreversible pulpitis, pulp necrosis, symptomatic apical periodontitis, chronic apical abscesses, acute apical abscesses without systemic involvement, tooth fractures, and concussions. These guidelines should be followed by dentists to reduce antibiotic overuse [1,7,8]. According to the World Health Organization (WHO) report published in 2014, inappropriate usage of antibiotics can lead to antimicrobial resistance (AMR) [9]. Although AMR is not a new phenomenon, it has recently become a serious issue as, under selective pressure due to the overuse of antibiotics, resistant clones are favored and the emergence and spread of resistance has rapidly accelerated [9]. Bacteria may be innately resistant to certain antibiotics, but may acquire antibiotic resistance through mutations in chromosomal genes and through horizontal gene transfer [9]. Thus, AMR significantly contributes to morbidity and mortality rates. For example, Prestinaci et al. estimated that about 400,000 infections and 25,000 deaths in Europe were caused by the most multi-drug resistant bacteria [10]. Given that in 2017, a 4.6% prevalence of intrahospital infections was reported by the Institute of Public Health of Serbia, it is essential to adopt more prudent antibiotic prescribing practices to avoid bacterial drug resistance [11]. A high level of resistance to all antibiotics tested, similar to that in the countries of southern and eastern Europe, has been observed in all tested bacterial species in the Republic of Serbia. The high level of resistance to carbapenems is of particular concern as these antibiotics are considered the last line of defense against antibiotic-resistant bacteria [12]. As a part of the implementation of the European Strategic Plan for Antibiotic Resistance, in 2017, the WHO for Europe established the Antimicrobial Medicines Consumption Network (AMC) to monitor the consumption of antimicrobial drugs among the AMC group members. According to the total consumption, the Republic of Serbia was in fourth place with 29.47 defined daily doses (DDDs) per 1000 individuals in 2014 [13]. This motivated the present study, as a part of which an overview of antibiotic prescribing practices for endodontic infections was conducted with the view of sharing the findings, along with the current ESE recommendations, with the dentists practicing in Serbia.

## 2. Materials and Methods

This research was approved by the Committee of Ethics of the Faculty of Medicine, University of Novi Sad (01-39/360/1, 11 December 2018). To obtain the data necessary for meeting the study objectives, an online questionnaire was developed and was made available to Serbian dentists via the following link: https://docs.google.com/forms/d/14J0BNQ6UWILc1cQCQoP4FSqIDmBLJ_Dd6xglALKfm-0. As the Law on Personal Data Protection (Legal Acts Republic of Serbia, number 87/2018) prohibits use of the Serbian Dental Chamber records for identifying study participants, only dentists whose email addresses were publicly available on the Internet were contacted and invited to take part in the survey. This resulted in an initial sample of 628 potential respondents. The survey items were adapted from previously published questionnaires with the permission of their authors [14,15]. The questionnaire commenced with a demographics section (gender, age, academic qualifications), followed by questions probing into the respondents’ antibiotic prescribing practices in relation to endodontic infections. To prevent duplicate data entries by the same individual, all respondents were required to provide their Serbian Dental Chamber identification number. To develop the questionnaire, a Google forms^®^ free online survey was utilized as it facilitated direct input of the survey data into a spreadsheet, thus aiding in subsequent analyses. Prior to its application, the questionnaire (Appendix A) was reviewed by the members of the endodontic section of the Department of Dentistry, Faculty of Medicine, University of Novi Sad. As one of the study goals was to promote compliance with the ESE recommendations for antibiotic use when treating endodontic infections, all survey respondents were provided with a link to the relevant files.

Statistical analyses were performed using SPSS^®^ 25.0 (SPSS, Inc., Chicago, IL, USA). Chi-squared test of independence was conducted to examine the potential relationships between nominal variables, with *p* < 0.05 considered statistically significant. In addition, Cramer’s V coefficient was calculated to assess the magnitude of correlations between nominal variables and as a measure of effect size when the Chi-squared test yielded statistically significant results.

## 3. Results

The link to the online survey was sent to all 628 dentists practicing in Serbia whose email addresses were publicly available, resulting in 158 (25.16%) completed questionnaires. According to surveysystem.com, this corresponded to a confidence interval of 6.75% at a 95% confidence level. The sample comprised of 92 female and 66 male dentists. Nearly half (48.7%) of the respondents were aged 36–45 years, while those aged 56–65 were the least represented (10.8%). When asked to indicate their educational attainment, 50.6% of the respondents indicated that they possessed a Doctor of Dental Medicine (DDM) undergraduate degree, 13.3% held a Master’s degree in Endodontics, and the remaining 36.1% held a master’s degree in other branches of dentistry. Further analyses revealed a positive relationship between academic degree and age (*p* = 0.009), which was expected, as specialization in any branch of dentistry is a time-consuming process. With regard to the duration of antibiotic treatment they recommended to their patients, most of the participating dentists stated that they prescribed a 5-day antibiotic course (Table 1).

A larger percentage of respondents who held Master’s degree in Endodontics prescribed a 7-day course of antibiotics compared to those with other qualifications, but this difference was not statistically significant.

For most respondents, Amoxicillin was the first-choice antibiotic for non-allergic patients. We, however, found that respondents aged 56–65 prescribed Amoxicillin + Clavulanic acid 1000 + 62.5 mg and Clindamycin 300 mg more often than dentists from other age groups, and this difference was statistically significant. On the other hand, dentists aged 46–55 prescribed Amoxicillin + Clavulanic acid 875 + 125 mg least frequently, while those aged 46–55 and 56–65 prescribed Azitromycin 500 mg more often than other age groups. While the aforementioned differences were statistically significant, no association between academic qualifications and prescribing practices were found (Table 2).

For patients that are allergic to penicillin, the majority of respondents prescribed Clindamycin. No relationship between academic qualifications or age and antibiotic prescribing practices for patients allergic to penicillin was found (Table 3).

When the participants’ responses related to antibiotic use in different clinical situations were analyzed, statistically significant differences emerged in relation to acute apical abscess with systemic involvement (Table 4). In particular, dentists aged 46–55 rarely prescribed antibiotics in these clinical situations.

## 4. Discussion

This was the first study in which Serbian dentists’ antibiotic prescribing practices for endodontic infections were examined quantitatively. As indicated earlier, pertinent data were obtained via an online survey comprised of survey items developed by other authors, which were replicated or adapted with their permission [14,15]. According to https://survey.com/, with 158 (25.16%) of the initially contacted dentists (628) responding to the survey, this sample size ensured a confidence interval of 6.75% at the 95% confidence level [14]. Unfortunately, despite sending two reminders, a greater response rate could not be achieved [16].

While the relatively small sample size is a notable limitation of this study, we were legally not permitted to reach out to the full population of Serbian dentists through the Serbian Dental Chamber. A further limitation stems from the self-reported nature of the data, as its veracity could not be established. Consequently, all study findings need to be interpreted with caution.

The majority of the respondents held a Doctor of Dental Medicine (DDM) undergraduate degree in dentistry, with a much smaller percentage of those having a master’s degree (either in endodontics or other branches of dentistry). As specialization in any branch of dentistry requires at least three years of additional education, as expected, older participants were more likely to hold a postgraduate degree.

In the treatment of endodontic infections, use of antibiotics is indicated in a small number of clinical situations, namely for treating acute apical abscesses in medically compromised patients, acute apical abscesses with systemic involvement, progressive infections, and persistent infections [1]. Considering that endodontic infections are usually characterized by rapid onset and short duration (2–7 days), when antibiotics are used, treatment duration of 3–7 days is often sufficient [3]. However, patients need to be seen 2–3 days after commencing antibiotic therapy to determine whether treatment should be stopped or continued [17]. In our survey, 55.7% of the respondents indicated that they typically prescribed a 5-day course of antibiotics, while 36.7% favored a 7-day course. Prescription period, however, was not correlated with the age or academic qualifications of respondents (Table 1).

Amoxicillin was the first-choice prescription for patients without an allergy to penicillin for 55.1% of the respondents, irrespective of age or academic qualifications (Table 2). This finding is in accordance with the results reported by other authors who conducted similar surveys in Europe [15,18,19,20,21,22]. As Amoxicillin is a β-lactam, moderate-spectrum antibiotic suited for treating oral infections [1], to increase its spectrum of action against *Staphilococcus aureus*, it is often prescribed in combination with clavulanic acid. In addition, β-lactam antibiotics (Amoxicillin, Penicillin V) are recommended by the European Society of Endodontology as the first option in the treatment of endodontic infections in non-allergic patients [3]. In Serbia, however, penicillin-containing drugs are marketed only as a benzylpenicillin powder for injections. For this reason, penicillin is not used by Serbian dentists, and was not included as one of the options in this survey.

According to the survey results, Clindamycin (600 mg) was the first-choice antibiotic in the treatment of endodontic infections in patients who are allergic to penicillin, irrespective of the dentist’s age and qualifications (Table 3). This practice is in agreement with the ESE recommendations [3], and concurs with the findings reported by other authors [14,15,20,23]. Clindamycin is a lincosamyde type of antibiotic with a wide spectrum of action and effective distribution in most body tissues [1]. Its concentration in bone is very similar to its plasma concentration [24]. According to the survey findings, for patients allergic to penicillin, most of the participants would prescribe Azithromycin and Erithromycin as the second choice (Table 3). Azithromycin belongs to the macrolide group of antibiotics with a wide spectrum of action and improved pharmacokinetics [25]. Erithromycin is also a macrolide antibiotic with a spectrum similar to that of penicillin, and is the first treatment choice for patients with an allergy to penicillin in India and Iran [26,27]. Unfortunately, Kuriyama et al. found that the *Fusobacterium* and *Prevotella* lineages from dentoalveolar infections were resistant to these antibiotics [28].

The present survey also inquired into the dentists’ antibiotic prescribing practices for various pulpal and periapical conditions (Table 4). According to the ESE recommendations, systemic use of antibiotics is necessary only for treating acute apical abscesses in medically compromised patients, acute apical abscesses with systemic involvement, progressive infections, and persistent infections [3]. Most importantly, antibiotic therapy is contraindicated in symptomatic irreversible pulpitis, pulp necrosis, acute apical periodontitis, chronic apical abscess, and acute apical abscess with no systemic involvement. According to the European Society of Endodontology report published in 2006, dental infections can be successfully treated by pulp extirpation, elimination of the source of infection, drainage, or tooth extraction [29]. In the present survey, only 1.3% of the respondents indicated that they relied on antibiotics in the treatment of symptomatic irreversible pulpitis (Table 4). This encouraging result is in accordance with contemporary attitudes and recommendations [3]. Similar percentages were reported by other authors: 2% by Skučaitė et al. [21], 4.4% by Mainjot et al. [19], 6.2% by Bolfoni et al. [14], and 7.4% by Peric et al. [23]. It is worth noting, however, that Tulip and Palmer and Rodriguez-Núñez reported 18% and 31.5%, respectively for England and Spain [18,20].

Even though it is known that antibiotics are ineffective in treating pulp necrosis [1], 3.2% of the survey respondents prescribed antibiotics in such cases (Table 4). Adequate endodontic treatment followed by three-dimensional obturation and coronal restoration are the correct and necessary clinical steps for the treatment of pulp necrosis [30]. Higher percentages of antibiotic use in such patients were found by Bolfoni et al. [14] and Segura Egea et al. [15] at 6.2% and 30.7%, respectively.

In cases of acute apical periodontitis with spontaneous pain, pain on percussion and biting, and widening of periodontal space, systemic antibiotic treatment is not required [1]. However, 12.7% of the survey respondents prescribed antibiotics as a treatment for these conditions (Table 4).

Similar percentage (11.5%) was reported by Bolfoni et al. [14], while Segura Egea et al. [15] reported a much higher percentage (71%). Root canal treatment is, in fact, the only recommended treatment for acute apical periodontitis [31,32] as well as for chronic apical abscesses (teeth with sinus tract, periapical radiolucency). However, these recommendations were not followed by 13.3% of our survey respondents (Table 4). According to this criterion, other investigators have reported a rather diverse prevalence of antibiotic use, namely Mainjot et al. [18] reported 2.7%, Bolfoni et al. [14] 20.5%, Rodriguez-Núñez et al. [19] 21.4%, Deniz-Sungur et al. [32] 26%, Nabavizadeh et al. [33] 58%, and Segura Egea et al. [15] reported 59.8%.

Even though acute apical abscesses with no systemic involvement characterized by localized fluctuant swelling do not require systemic usage of antibiotics, but rather root canal treatment [1], 31% of the survey respondents prescribed antibiotics in this situation (Table 4). Of course, this practice should stop, as antibiotic overuse may lead to antimicrobial resistance. Unfortunately, similar findings were reported by other authors: 51.9% by Mainjot et al. [18], 52.9% by Rodriguez-Núñez et al. [19], 71% by Segura Egea et al. [15], 71.5% by Bolfoni et al. [14], and 74.2% by Nabavizadeh et al. [33].

In acute apical abscesses in medically compromised patients (immunocompromised patients, patients with *locus minoris resistentiae*), antibiotic use is indicated because systemic diseases result in impaired immunologic function [1]. In the present survey, 87.3% of the respondents confirmed that they prescribed antibiotics in these cases (Table 4). No links between dentist’s age and qualifications with the prescribing pattern could be established. In other surveys, antibiotic prophylaxis for medically compromised patients is rarely explored, due to which no comparisons with the findings of other authors can be made. However, this practice is in line with the ESE criteria [3].

Antibiotics are essential in the treatment of acute apical abscesses with systemic involvement [3]; hence, it is encouraging that 96.2% of the respondents in our study adhered to these guidelines (Table 4). However, dentists aged 46–55 were statistically significantly less likely than those in other age groups to have prescribed antibiotics in this situation. Namely, only 88.2% of respondents in this age group prescribed antibiotics for acute apical abscesses with systemic involvement, while almost all respondents in other age groups prescribed antibiotics for treating this condition (Table 4). These results are comparable with those reported by Bolfoni et al. [14], Rodriguez-Núñez et al. [19], and Segura Egea et al. [15] at 88.1%, 94.3%, and 94.5%, respectively.

According to the survey findings, 79.7% of the respondents utilized systemic antibiotic therapy when treating progressive infections (Table 4), and this is in line with the ESE recommendations [3]. Progressive infection is characterized by a rapid onset of severe infection (within 24 h), cellulitis or a spreading infection, and osteomyelitis, and thus necessitates systemic antibiotic therapy.

Persistent infection, manifesting as chronic exudation that is not resolved by regular intracanal procedures and medications, also requires antibiotic treatment [3]. However, only 59.5% of the survey respondents prescribe antibiotics in such cases (Table 4). This may suggest failure to keep abreast of the current recommendations and lack of postgraduate training. As one of the aims of the present study was disseminating the ESE recommendations to the dentists who completed the questionnaire, this initiative may aid in mitigating these deficiencies.

For patients experiencing postoperative pain, antibiotics are not indicated; however, they are prescribed by 6.3% of the survey respondents (Table 4). Postoperative pain is one of the sequelae that may discourage patients from pursuing root canal therapy. A slightly lower percentage (4.9%) was reported by Bolfoni et al. [14], most likely due to the fact that their survey specifically targeted Brazilian endodontists, while the majority of the respondents in our study were dentists.

It is encouraging that, according to this survey, the majority of dentists in Serbia prescribe antibiotics in clinical situations in which this is warranted. It is also noteworthy that a small percentage of respondents prescribed antibiotics for treating irreversible pulpitis, pulp necrosis, acute and chronic apical periodontitis, and acute apical abscess with no systemic involvement (Table 4). On the other hand, although antibiotics are frequently prescribed in cases of progressive infections, the percentage should be higher.

Finally, it is reassuring to note that only 1.3% of our survey participants routinely prescribed antibiotics for patients undergoing endodontic treatment, irrespective of the diagnosis (Table 4).

Hence, raising the awareness of Serbian dentists of correct antibiotic use in the treatment of endodontic infections is an important step in the global fight against AMR. Such information dissemination should start during their undergraduate training and regular updates should be made available to all practicing dentists. In undergraduate education, the severity of the AMR phenomenon should be emphasized and the guidelines for appropriate antibiotic use should be frequently reiterated. Similarly, conferences and seminars, along with online repositories, are ideal platforms for ongoing education of dental practitioners at all levels.

## 5. Conclusions

Despite a relatively small sample size and the self-reported nature of the data analyzed as a part of this investigation, it can be tentatively concluded that a significant percentage of Serbian dentists prescribe antibiotics responsibly. Unfortunately, recommendations are not always followed, and in certain cases, patients are given antibiotics even when this is not indicated. The disparity between the actual and recommended prescribing practices supports the need for additional education on responsible antibiotic use. We hope that by distributing the ESE guidelines to the study participants, we have contributed to this ongoing endeavor.

## Figures and Tables

**Table 1 antibiotics-10-00067-t001:** Duration of antibiotic treatment according to the respondents’ age and academic qualifications.

Treatment Duration	Age (Years)				Academic Qualifications			Total
	25–35	36–45	46–55	56–65	Doctor of Dental Medicine (DDM) Undergraduate Degree	Master’s Degree in Other Branch of Dentistry	Master’s Degree in Endodontics	
3 days	1 (3.3%)	1 (1.3%)	3 (8.8%)	1 (5.9%)	1 (1.3%)	4 (7%)	1 (4.8%)	3.8%
5 days	13 (43.3%)	43 (55.8%)	22 (64.7%)	10 (58.8%)	45 (56.3%)	34 (59.6%)	9 (42.9%)	55.7%
7 days	16 (53.3%)	29 (37.7%)	7 (20.6%)	6 (35.3%)	32 (40%)	16 (28.1%)	10 (47.6%)	36.7%
10 days	0 (0%)	2 (2.6%)	1 (2.9%)	0 (0%)	2 (2.5%)	1 (1.8%)	0 (0%)	1.9%
Until symptoms disappear	0 (0%)	2 (2.6%)	1 (2.9%)	0 (0%)	0 (0%)	2 (3.5%)	1 (4.8%)	1.9%
*p*-value	0.297				0.294			
Cramer’s V	0.161				0.174			

**Table 2 antibiotics-10-00067-t002:** First-choice antibiotics according to the participants’ age and academic qualifications (statistically significant differences are highlighted in red).

Antibiotic	Age (Years)				Academic Qualifications			Total
	25–35	36–45	46–55	56–65	DDM Undergraduate Degree	Master’s Degree in Other Branch of Dentistry	Master’s Degree in Endodontics	
Amoxicilin + Orvagil 500 + 400 mg	0 (0%)	2 (2.6%)	0 (0%)	0 (0%)	2 (2.5%)	0 (0%)	0 (0%)	1.3%
Amoxicillin 1000 mg	0 (0%)	4 (5.2%)	0 (0%)	0 (0%)	1 (1.3%)	3 (5.3%)	0 (0%)	2.5%
Amoxicillin 250 mg	1 (3.3%)	3 (3.9%)	1 (2.9%)	0 (0%)	1 (1.3%)	4 (7%)	0 (0%)	3.2%
Amoxicillin 500 mg	20 (66.7%)	40 (51.9%)	20 (58.8%)	7 (41.2%)	45 (56.3%)	32 (56.1%)	10 (47.6%)	55.1%
Amoxicillin + Metronidazol 500 + 400 mg	0 (0%)	1 (1.3%)	0 (0%)	0 (0%)	1 (1.3%)	0 (0%)	0 (0%)	0.6%
Amoxicillin + Clavulanic acid 1000 + 62.5 mg	0 (0%)	0 (0%)	0 (0%)	2 (11.8%)	2 (2.5%)	0 (0%)	0 (0%)	1.3%
Amoxicillin + Clavulanic acid 500 + 125 mg	2 (6.7%)	7 (9.1%)	1 (2.9%)	0 (0%)	3 (3.8%)	4 (7%)	3 (14.3%)	6.3%
Amoxicillin + Clavulanic acid 875 + 125 mg	1 (3.3%)	5 (6.5%)	0 (0%)	2 (11.8%)	3 (3.8%)	4 (7%)	1 (4.8%)	5.1%
Azithromycin 250 mg	0 (0%)	2 (2.6%)	0 (0%)	0 (0%)	2 (2.5%)	0 (0%)	0 (0%)	1.3%
Azithromycin 500 mg	0 (0%)	0 (0%)	1 (2.9%)	1 (5.9%)	0 (0%)	2 (3.5%)	0 (0%)	1.3%
Clindamycin 300 mg	0 (0%)	0 (0%)	0 (0%)	2 (11.8%)	1 (1.3%)	0 (0%)	1 (4.8%)	1.3%
Clindamycin 600 mg	6 (20%)	11 (14.3%)	9 (26.5%)	3 (17.6%)	16 (20%)	7 (12.3%)	6 (28.6%)	18.%
Dovicin 100 mg	0 (0%)	2 (2.6%)	1 (2.9%)	0 (0%)	3 (3.8%)	0 (0%)	0 (0%)	1.9%
Depending on the indication	0 (0%)	0 (0%)	1 (2.9%)	0 (0%)	0 (0%)	1 (1.8%)	0 (0%)	0.6%
*p*-value	0.009				0.369			
Cramer’s V	0.365				0.314			

**Table 3 antibiotics-10-00067-t003:** Antibiotics of choice for patients allergic to penicillin according to the participants’ age and academic qualifications.

Antibiotic	Age (Years)				Academic Qualifications			
	25–35	36–45	46–55	56–65	DDM Undergraduate Degree	Master’s Degree in Other Branch of Dentistry	Master’s Degree in Endodontics	Total
Azithromycin	2 (6.7%)	14 (18.2%)	5 (14.7%)	2 (11.8%)	12 (15%)	11 (19.3%)	0 (0%)	14.6%
Clindamycin	21 (70%)	43 (55.8%)	21 (61.8%)	12 (70.6%)	46 (57.5%)	32 (56.1%)	19 (90.5%)	61.4%
Dovicin	0 (0%)	1 (1.3%)	0 (0%)	0 (0%)	1 (1.3%)	0 (0%)	0 (0%)	0.6%
Doxiciklin	0 (0%)	1 (1.3%)	0 (0%)	0 (0%)	1 (1.3%)	0 (0%)	0 (0%)	0.6%
Erithromycin	7 (23.3%)	11 (14.3%)	3 (8.8%)	2 (11.8%)	14 (17.5%)	7 (12.3%)	2 (9.5%)	14.6%
Metronidazole	0 (0%)	4 (5.2%)	3 (8.8%)	1 (5.9%)	4 (5%)	4 (7%)	0 (0%)	5.1%
Roxitromicin	0 (0%)	1 (1.3%)	0 (0%)	0 (0%)	0 (0%)	1 (1.8%)	0 (0%)	0.6%
Tetracycline	0 (0%)	2 (2.6%)	1 (2.9%)	0 (0%)	2 (2.5%)	1 (1.8%)	0 (0%)	1.9%
Depending on the indication	0 (0%)	0 (0%)	1 (2.9%)	0 (0%)	0 (0%)	1 (1.8%)	0 (0%)	0.6%
*p*-value	0.885				0.461			
Cramer’s V	0.184				0.224			

**Table 4 antibiotics-10-00067-t004:** Antibiotic use for different clinical indications according to the participants’ age and academic qualifications (statistically significant differences are highlighted in red).

Diagnosis	Age				*p*-Value	Cramer’s	Academic	Qualifications		*p*	Cramer’s V	Total
	25–35	36–45	46–55	56–65		V	DDM Undergraduate Degree	Master’s Degree in Other Branch of Dentistry	Master’s Degree in Endodontics			
Symptomatic Irreversible pulpitis	0 (0%)	1 (1.3%)	1 (2.9%)	0 (0%)	0.713	0.093	2 (2.5%)	0 (0%)	0 (0%)	0.373	0.112	1.3%
Pulp necrosis	0 (0%)	5 (6.5%)	0 (0%)	0 (0%)	0.143	0.185	4 (5%)	1 (1.8%)	0 (0%)	0.380	0.111	3.2%
Acute apical periodontitis	5 (16.7%)	11 (14.3%)	3 (8.8%)	1 (5.9%)	0.620	0.106	11 (13.8%)	8 (14%)	1 (4.8%)	0.505	0.093	12.7%
Chronic apical abscess	4 (13.3%)	9 (11.7%)	6 (17.6%)	2 (11.8%)	0.858	0.070	13 (16.3%)	7 (12.3%)	1 (4.8%)	0.371	0.112	13.3%
Acute apical abscess with no systemic involvement	12 (40%)	27 (35.1%)	8 (23.5%)	2 (11.8%)	0.135	0.188	23 (28.8%)	23 (40.4%)	3 (14.3%)	0.072	0.182	31%
Acute apical abscess in medically compromised patients	27 (90%)	70 (90.9%)	26 (76.5%)	15 (88.2%)	0.193	0.173	68 (85%)	52 (91.2%)	18 (85.7%)	0.542	0.088	87.3%
Acute apical abscess with systemic involvement	29 (96.7%)	76 (98.7%)	30 (88.2%)	17 (100%)	0.048	0.224	75 (93.8%)	56 (98.2%)	21 (100%)	0.247	0.133	96.2%
Progressive infections	22 (73.3%)	66 (85.7%)	25 (73.5%)	13 (76.5%)	0.336	0.146	60 (75%)	47 (82.5%)	19 (90.5%)	0.238	0.135	79.7%
Persistent infections	16 (53.3%)	46 (59.7%)	20 (58.8%)	12 (70.6%)	0.718	0.092	46 (57.5%)	36 (63.2%)	12 (57.1%)	0.780	0.056	59.5%
Post-operative pain	2 (6.7%)	6 (7.8%)	1 (2.9%)	1 (5.9%)	0.814	0.077	7 (8.8%)	2 (3.5%)	1 (4.8%)	0.440	0.102	6.3%
During endo treatment	0 (0%)	1 (1.3%)	1 (2.9%)	0 (0%)	0.713	0.093	2 (2.5%)	0 (0%)	0 (0%)	0.373	0.112	1.3%

## Data Availability

Data available in a publicly accessible repository.

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
