# Peer review of "Antibiotic Prescribing Practices in Endodontic Infections: A Survey of Dentists in Serbia"

_antibiotics, 2021, doi:10.3390/antibiotics10010067_

Round 1
Reviewer 1 Report
In this manuscript the authors performed a survey on the prescribing practices of antibiotics in dentistry. A cohort of 158 Serbian dentists was investigated: the results showed that 55.7% of dentists prescribed a 5-day antibiotic therapy for endodontic infections. Amoxicillin was the first-choice antibiotic (55.1%), followed by Clindamycin (18.4%). Furthermore, for patients that are allergic to penicillin, 61.4% of respondents prescribe Clindamycin. The study is on a timely subject since the antibiotic resistance has emerged as one of the principal public health problems of the 21st century. As a reviewer of this manuscript, I object to a number of comments about the design of this study.
Abstract: please insert the number of questionnaires analyzed (n = 158) and not only the number of initial potential responders (n = 628).
Results: Delete the percentages (if n < 100, the use of percentage implies a spurious impression of accuracy).
In different parts of the text the Authors use the term “correlation”. I understand that the Authors want to speak to dependence, association or relationship, once correlation implies in a linear relationship between two variables. I suggest using other terms, such as “relationship” or “association”.
Lines 158-163: move the comments on the results obtained in the Discussion section.
Discussion: move the tables in the Results section and insert a legend to explain the meaning of the numbers in red.
Furthermore, minor language corrections should be necessary.
Author Response
Response to Reviewer 1 Comments
Point 1: Abstract: please insert the number of questionnaires analyzed (n = 158) and not only the number of initial potential responders (n = 628).

Response 1: Thank you for such a beneficial suggestion. In response, we have now provided the number of analyzed questionnaires in the Abstract.
Point 2: Results: Delete the percentages (if n < 100, the use of percentage implies a spurious impression of accuracy).
Response 2: Thank you for pointing this put. In accordance with your suggestion, we have deleted the percentages in all cases where n < 100 applies.
Point 3. Results. In different parts of the text the Authors use the term “correlation”. I understand that the Authors want to speak to dependence, association or relationship, once correlation implies in a linear relationship between two variables. I suggest using other terms, such as “relationship” or “association”.
Response 3: Thank you for this valuable suggestion. In the revised manuscript, we have replaced all instances of “correlation” with the proposed terms.
Point 4. Results. Lines 158-163: move the comments on the results obtained in the Discussion section.
Response 4: We appreciate this observation. Accordingly, we have removed the indicated comments from the Discussion section.
Point 5. Discussion: move the tables in the Results section and insert a legend to explain the meaning of the numbers in red.
Response 5: Thank you for such valuable feedback. To address this comment, we have moved the indicated tables and have created legends for each one.
We certify that the manuscript has been thoroughly revised by a British professional academic editor with a PhD in nuclear biophysics (specializing in medical imaging) gained at University of London, Faculty of Medicine, with more than 10 years of experience editing academic articles and PhD dissertations. If needed, we will provide full details of her academic qualifications along with her contact details.
Reviewer 2 Report
Authors surveyed the knowledge and attitudes of Serbian dentists in antibiotic prescribing when treating endodontic infections.
The manuscript has so many typos and grammatical errors and an English check by academic professional is a must.
The abstract contains abbreviations that has not been outlined, such as ESE. The acronym should been written fully in their first appearance in the text.
Abstract:
1- authors stated "Statistical analysis was performed using SPSS® 25.0 (SPSS, Inc., Chicago, IL, USA) and Chi-squared test and Cramer’s V measure of association (p<0.05) were performed when testing possible associations." This is very confusing. The methods that are used in the analysis is different to the software. please rewrite.
2- the results in the abstract should report the overall response rate (25.16%).
Introduction:
there are similar studies that have been undertaken recently and authors can enhance their introduction and discussion sections by including them:
Deniz-Sungur D, Aksel H, Karaismailoglu E, Sayin TC. The prescribing of antibiotics for endodontic infections by dentists in Turkey: a comprehensive survey. Int Endod J. 2020 Dec;53(12):1715-1727. doi: 10.1111/iej.13390. Epub 2020 Sep 18. PMID: 32805741.
Mende A, Venskutonis T, Mackeviciute M. Trends in Systemic Antibiotic Therapy of Endodontic Infections: a Survey among Dental Practitioners in Lithuania. J Oral Maxillofac Res. 2020 Mar 31;11(1):e2. doi: 10.5037/jomr.2020.11102. PMID: 32377326; PMCID: PMC7191382.
Methods:
how was the questionnaire was developed? has this been used from a previous study? if so, please refer and cite accordingly. Has the questionnaire been circulated by the Serbian language? if the questionnaire was translated from English to Serbian, how the translation been validated?
I have a major concern about the authors statement on how calculating the sample size. 25 is not reasonable???
how many reminders the authors sent to improve the response rate?
Figure 1 is not useful and can be moved as a supplemental material. the quality of tables should be improved by adding more details to the lines, columns and the table legends.
in tables 2 and 4, there are some data highlighted in red, why?
The results are mostly described in the discussion rather in the results section. please rewrite both sections.
the discussion should focus also how to improve the current practices of dentists in Serbia regarding prescribing antibiotics.
Author Response
Response to Reviewer 2 Comments
Point 1: authors stated "Statistical analysis was performed using SPSS® 25.0 (SPSS, Inc., Chicago, IL, USA) and Chi-squared test and Cramer’s V measure of association (p<0.05) were performed when testing possible associations." This is very confusing. The methods that are used in the analysis is different to the software. please rewrite. 

Response 1: Thank you for this valuable suggestion. Accordingly, we have rewritten this part of Abstract to ensure that the analysis methods mentioned align with those discussed in the main paper.
Point 2: the results in the abstract should report the overall response rate (25.16%).
Response 2: We really appreciate this observation and have accordingly provided the response rate in the Abstract.
Point 3: Introduction:
there are similar studies that have been undertaken recently and authors can enhance their introduction and discussion sections by including them:
Deniz-Sungur D, Aksel H, Karaismailoglu E, Sayin TC. The prescribing of antibiotics for endodontic infections by dentists in Turkey: a comprehensive survey. Int Endod J. 2020 Dec;53(12):1715-1727. doi: 10.1111/iej.13390. Epub 2020 Sep 18. PMID: 32805741.
Mende A, Venskutonis T, Mackeviciute M. Trends in Systemic Antibiotic Therapy of Endodontic Infections: a Survey among Dental Practitioners in Lithuania. J Oral Maxillofac Res. 2020 Mar 31;11(1):e2. doi: 10.5037/jomr.2020.11102. PMID: 32377326; PMCID: PMC7191382.
Response 3: We really appreciate it that you took the time to identify relevant literature sources for inclusion in the Introduction and Discussion sections. We have now cited these articles in the main text and have included them into the list of references. Specifically, Mende A, Venskutonis T, Mackeviciute M. Trends in systemic antibiotic therapy of endodontic infections: A survey among dental practitioners in Lithuania. J Oral Maxillofac Res. 2020 Mar 31;11(1):e2. doi: 10.5037/jomr.2020.11102. PMID: 32377326; PMCID: PMC7191382, is included under number [8] and Deniz-Sungur D, Aksel H, Karaismailoglu E, Sayin TC. The prescribing of antibiotics for endodontic infections by dentists in Turkey: A comprehensive survey. Int Endod J. 2020 Dec;53(12):1715-1727. doi: 10.1111/iej.13390. Epub 2020 Sep 18. PMID: 32805741 is included under number [32].
Point 4: Methods:
how was the questionnaire was developed? has this been used from a previous study? if so, please refer and cite accordingly. Has the questionnaire been circulated by the Serbian language? if the questionnaire was translated from English to Serbian, how the translation been validated?
Response 4: Thank you for seeking this clarification. Indeed, the questionnaire utilized in our search was originally developed by Bolfoni MR, Pappen FG, Pereira-Cenci T, and Jacinto RC (Antibiotic prescription for endodontic infections: A survey of Brazilian endodontists. Int Endod J 2018;51: 148–156) and Segura-Egea JJ, Velasco-Ortega E, Torres-Lagares D, Velasco-Ponferrada MC, Monsalve-Guil L, and LLamas-Carreras JM (Pattern of antibiotic prescription in the management of endodontic infections among Spanish oral surgeons. Int Endod J 2010;43: 342–350) and was adopted with the permission from the authors. For the purposes of data collection in Serbia, the original questionnaire was translated to Serbian language and the translation was validated by academic professionals prior to distributing it to the study participants.
Point 5: Methods:
I have a major concern about the authors statement on how calculating the sample size. 25 is not reasonable???
how many reminders the authors sent to improve the response rate?
Response 5: Thank you for raising this concern. Our original manuscript unfortunately contained a calculation error, as for a 4% confidence interval and a 95% confidence level, a minimum sample size is 307. Consequently, for our sample of 158 respondents, at the 95% confidence level, the confidence interval is 6.75%. This mistake has now been rectified, but we can delete the relevant sentences if you agree with this approach. Regarding your second question, two reminders were sent to the dentists.
Point 6: Methods:
Figure 1 is not useful and can be moved as a supplemental material. the quality of tables should be improved by adding more details to the lines, columns and the table legends.
in tables 2 and 4, there are some data highlighted in red, why?
The results are mostly described in the discussion rather in the results section. please rewrite both sections.
the discussion should focus also how to improve the current practices of dentists in Serbia regarding prescribing antibiotics.
Response 6:
Thank you for making such beneficial suggestions. In response, we have removed Figure 1 from the revised manuscript and have provided table legends, as indicated. We also offer an explanation for highlighting some of the results reported in Table 2 and 4 in red. These were the only results that reached statistical significance and we wanted them to stand out.
We have also revised both the Results and the Discussion section, as requested. In response to your last comment, we have now added a paragraph on the value of our study for improving the current antibiotic prescribing practices of dentists in Serbia.
We certify that the manuscript has been thoroughly revised by a British professional academic editor with a PhD in nuclear biophysics (specializing in medical imaging) gained at University of London, Faculty of Medicine, with more than 10 years of experience editing academic articles and PhD dissertations. If needed, we will provide full details of her academic qualifications along with her contact details.
Round 2
Reviewer 2 Report
Authors responded to my comments by providing a letter but they have not made the required changes to the manuscript. 90% of my comments have not been addressed properly. These comments to improve the clarity of your manuscript. Regrettably, I can't endorse the current version for publication.
The manuscript doesn't flow and read well and the language still warrants major revision.
The abstract can be improved substantially by adding more details to the results. such statement "Statistical analyses were performed using SPSS® 25.0 (SPSS, Inc., Chicago, IL, USA)." should be deleted.
The details about the source of the questionnaire, translation into Serbian language, and validating the questionnaire afterwards should be added to the methods section. Also, authors should acknowledge the original articles that have used to build up their questionnaire.
Author Response
Reply to Academic Editor
Dear Sir,
Thank You for reviewing of our manuscript and giving us a large number of beneficial suggestions. We changed our manuscript according to these suggestions, and hope that this new version is available for Antibiotics journal.
Sincerely,
Prof. dr Milan Drobac